# Effects of Dietary Substitution of Alfalfa Silage with Virginia Fanpetals Silage in Lactating Polish Holstein Friesian Dairy Cows

**DOI:** 10.3390/ani10101746

**Published:** 2020-09-25

**Authors:** Cezary Purwin, Zenon Nogalski, Maciej Starczewski, Sylwia Czurgiel, Maja Fijałkowska, Martyna Momot, Marta Borsuk

**Affiliations:** 1Department of Animal Nutrition and Feed Science, University of Warmia and Mazury in Olsztyn, Oczapowskiego 5, 10-719 Olsztyn, Poland; Maciej_Starczewski@cargill.com (M.S.); sylwia.kotlarczyk@uwm.edu.pl (S.C.); maja.fijalkowska@uwm.edu.pl (M.F.); marta.borsuk@uwm.edu.pl (M.B.); 2Department of Cattle Breeding and Milk Evaluation, University of Warmia and Mazury in Olsztyn, Oczapowskiego 5, 10-719 Olsztyn, Poland; zenon.nogalski@uwm.edu.pl (Z.N.); martyna.momot@uwm.edu.pl (M.M.)

**Keywords:** Virginia fanpetals silage, *Sida hermaphrodita* silage, dairy cows, milk quality, milk composition, alfalfa substitution

## Abstract

**Simple Summary:**

Alfalfa silage, owing to its chemical composition, complements maize silage in many rations for dairy cows. The disadvantages of alfalfa are the high soil requirements, agrotechnical treatments and sensitivity to cold and flooding. Many dairy farms are unable to grow alfalfa because of poor quality of the soils, and grass crops are unreliable due to increasing drought. For this reason, alternative plants are being sought out to produce feeds for dairy cows. So far, Virginia fanpetals has been used as an energy source, but thanks to its high levels of protein (17–25%), the possibility of using it in cattle feed has been recognized. Virginia fanpetals grows well in poor quality soils, in difficult climatic conditions (frost, drought), with no need for annual treatments, and the harvest is carried out with maize harvesting equipment. To date, no studies have been carried out on the possibility of replacing alfalfa with Virginia fanpetals in the diets of dairy cows. Based on our results, it was concluded that Virginia fanpetals is a good source of protein and can be a substitute or supplement to alfalfa silage. The best production results were obtained with half-substitution of alfalfa silage by Virginia fanpetals silage in diets based on maize silage.

**Abstract:**

The aim of this study was to evaluate the effect of partial or complete substitution of alfalfa silage with Virginia fanpetals silage in rations based on maize silage on feed intake, digestibility, ruminal fermentation and milk yield and physicochemical characteristics. Nine Polish Holstein Friesian cows in the second half of lactation were fed three experimental diets in a replicated 3 × 3 Latin square design as follows: maize silage + alfalfa silage, maize silage + alfalfa silage and Virginia fanpetals silage in a 50:50 ratio, maize silage + Virginia fanpetals silage. Complete substitution caused an increase in dry matter intake (DMI), total volatile fatty acids (VFA), acetic acid to propionic acid (A/P) ratio, N-NH_3_ in the rumen contents and milk urea and a decrease in the feed conversion ratio. The partial and complete substitution changed the profile of milk fatty acids, resulting in a slight increase in saturated fatty acids (SFA) and a decrease in unsaturated fatty acids (UFA) as well as in all functional fatty acids except vaccenic acid. The most promising production effects were achieved through partial substitution of alfalfa silage with the Virginia fanpetals silage.

## 1. Introduction

One of the principal challenges in the nutrition of high-yielding dairy cows is to supplement maize silage with proteins while maintaining the high density of a ration and limited consumption of expensive protein supplements. Alfalfa silage (*Medicago sativa* L.) is a popular bulk feed possessing nutritional value which is complementary to that of maize [1,2]. Ensiled alfalfa is used in cattle nutrition owing to its high content of crude protein (CP) and low share of neutral detergent fiber (NDF), in addition to which it can maximize feed intake and milk yield [3].

The major advantages of alfalfa are the high yield of dry matter (DM) and CP, the fact that it grows and yields for several years, and that it is tolerant to droughts and high temperatures [4]. However, alfalfa must be grown on soils that are fertile, brittle, dried, loamy and rich in nutrients; the crop also requires several agronomic treatments, such as ploughing, fertilization and liming. Moreover, alfalfa is sensitive to cold weather in spring and to periodic flooding [5]. In Poland, many dairy farms are in areas where the quality of soils, provided intensive organic fertilization is undertaken, is suitable for maize but precludes the cultivation of alfalfa, while yields of grasses are unreliable due to increasingly frequent droughts. It is therefore necessary to look for alternative plants for the production of fodder on sandy, degraded or marginal soils, and which may supplement maize in feeding rations [6]. Virginia fanpetals (*Sida hermaphrodita L. Rusby*), a perennial plant from the family *Malvaceae*, originating from North America, grows under the soil and climate conditions present in central Europe and has been proven to be an efficient source of biomass. Virginia fanpetals can grow on a plantation for up to 20 years, generating high yields of biomass (from 9 to 20 t/ha DM) while presenting low nutritional requirements. The plant first grows to a height of 1.2–1.8 m, while the oldest plants can reach over 4 m in height [7,8,9,10]. Unlike alfalfa, Virginia fanpetals can be successfully grown on sandy or rocky soils which are poor in organic matter and nutrients, where the crop simultaneously minimizes soil erosion and improves the structure and fertility of the soil [11]. Virginia fanpetals is highly tolerant to low temperatures (to −35 °C), drought and lodging [12,13], while not being sensitive to soil pH [14]. The highest yields of biomass are harvested at the budding phase, when the plant contains 15–22% DM and 17–25% of CP [13]. The fact that Virginia fanpetals grows long and yields high means that the CO_2_ emissions involved in its cultivation are lower, as some annual treatments (ploughing, harrowing) are redundant. Moreover, biomass can be harvested in a conventional way, using the equipment for maize harvest [15] and not incurring high financial outlays [16]. Finally, Virginia fanpetals offers ecological benefits owing to its the ability to store carbon in its well-developed root system, consequently keeping it sequestered underground for many years [17].

Currently, the cultivation of Virginia fanpetals is gaining popularity in European countries [16]. Thus far, it has been grown mainly as an alternative source of energy, in which case the plants are harvested later, for dry stems, which are then processed into pellets or burnt as chips [6,10,18]. Studies on the biomass of Virginia fanpetals have shown that it differs from the those of other energy crops in its higher CP content. Given its ability to regrow after cutting during the plant growing season, Virginia fanpetals has been considered as a source of cosubstrate for maize silage in agricultural biogas plants [13]. The verified potential of this species for methane production owing to its high yield implicates good microbial decomposition of its organic substances [19]. Because the chemical composition of alfalfa resembles that of Virginia fanpetals, it is suspected that similar production outputs will be achievable when alfalfa silage is substituted with Virginia fanpetals silage in rations based on maize silage.

The objective of this study was to determine the effect of partial or complete substitution of alfalfa silage with Virginia fanpetals silage in the diets of lactating Polish Holstein Friesian cows on the intake and digestibility of rations, ruminal fermentation and milk yield and physicochemical characteristics.

## 2. Materials and Methods

The experimental animals were maintained in compliance with the requirements specified in the Directive 2010/63/EU of the European Parliament and of the Council on the Protection of Animals Used for Scientific Purposes [20]. The experiment was conducted on a commercial farm (in the Province of Warmia and Mazury, Poland). Chemical analyses were made in a laboratory at the University of Warmia and Mazury in Olsztyn (Poland).

### 2.1. Silages

The biomass of Virginia fanpetals (*Sida hermaphrodita L. Rusby*) was collected from a plantation in the fifth year of its use, situated in northern Poland, on sandy soil fertilized with N80 P20 K60 kg/ha. The harvest was carried out in the plant budding phase, on 16 June, 2016, between 12:00 and 2:00 p.m. The plants were harvested by direct cut at a height 20 cm, using a self-propelled forage harvester Claas Jaguar 870 Profi, Kemper 445 (GmbH, Harsewinkel, Germany) with a cracker. The theoretical length of particles was 18 mm. The biomass was piled in a heap on a concrete plate covered with a 20-cm thick layer of straw, while the sidewalls consisted of stacked straw bales. Alfalfa silage was produced from plants harvested on a two-year-old plantation. The plants were cut after 32 days of regrowth using a mower (CLAAS corto 270, GmbH), and collected 24 h later with a precision chop forage harvester (John Deere 7050, Deere & Company, Moline, IL, USA). Maize silage was made from the hybrid Ronaldinio (KWS, Poznań, Poland), harvested in the dough stage, on 30 September 2016. Alfalfa silage and maize silage were compacted in horizontal silos. All three silages were compressed using a telehandler Manitou MLT 735 PowerShift (Manitou BF, Ancenis, France) and then covered with underlay silage film 40 µm in thickness (Bag Polska Sp. z o.o., Krzemieniewo, Poland), three-layer polyethylene film SILO-VIT^®^M-Silo 120 µm in thickness (RKW Hyplast, Hoogstraten, Belgium) and a silage cover.

### 2.2. Cows, Experimental Design and Diets

Nine Polish Holstein Friesian dairy cows were used in the experiment. At the onset of the trial, the cows were at day 105 of lactation on average, and their mean daily milk yield was 28.14 kg; the average body weight was 605 ± 24 kg and 612 ± 31.7 kg at the beginning and at the end of the experiment, respectively. The design followed a triple Latin square (3 × 3). The cows were assigned to each square according to the method of analogues, taking into account a sequence of three experimental diets during each of the three 30-day periods and 15 days of adaptation. The three experimental diets consisted of maize silage and concentrate, as well as alfalfa silage (Alfalfa 100%), alfalfa silage and Virginia fanpetals silage in a 50:50 ratio (Alfalfa 50%:Virginia fanpetals 50%), and Virginia fanpetals silage (Virginia fanpetals 100%) (Table 1). The bulk feed to concentrate ratio was 70:30 according to DM. The share of CP from alfalfa silage and Virginia fanpetals silage to the total CP in the three diets was, respectively, 455, 441 and 445 g/kg. The following values were calculated using the chemical composition of forages and in vitro dry matter digestibility (IVDMD), based on an equation presented by Jarrige [21]: feed unit for lactation (UFL), protein digested in the small intestine when nitrogen is limiting (PDIN), protein digested in the small intestine when energy is limiting (PDIE) and fill unit for dairy cows (FUC). Isonitrogenous diets (15.6% CP on average) were designed according to the recommendations of INRA for net energy (NE), PDIN, PDIE for a dairy cow with average lactation, weighing 650 kg and producing 26.0 kg milk/d containing 4.4% fat and 19.2 kg/d DMI [22]. Rations were prepared once daily at 6:00 a.m. by mixing the ingredients for about 10 min in a feeding cart Sano TMR Profi Kompakt 5 m^3^ (Sano-Modern Animal Nutrition Sp. z o.o., Sękowo, Poland), after which appropriate batches were weighed. The mixed and weighed rations were fed to cows twice daily: at 7:00 a.m. in an amount of 70% of the ration weight, and at 3:00 p.m., comprising the remaining 30% of the daily feeding ration. The weight of a ration was intended to prevent the cows from leaving more than 5–10% of uneaten fodder. The refusals were collected and weighed for 7 consecutive days. The cows were maintained in a tied-up system, on rubber mats, and fed individually with ad libitum access to water. The cows were milked in their standings, twice daily, at 5:00 a.m. and 5:00 p.m., using a mobile milking machine. The milk obtained from the cows was weighed to 0.01 kg precision.

### 2.3. Milk Sampling

Samples of milk for analyses were collected separately from each cow three times a month, after the morning milking. The milk samples were delivered to the laboratory of the Department of Cattle Breeding and Milk Evaluation at the University of Warmia and Mazury in Olsztyn (Poland). The following were determined in every milk sample: active acidity (pH) with a pH-meter by Hanna Instruments HI 99,163 (Hanna Instruments, Woonsocket, RI, USA), density with an electronic densimeter DMA 35N (Anton Paar, GmbH, Graz, Austria), content of fat, protein, lactose, dry matter, fat-free dry matter, urea, and total acidity on a MilkoScan apparatus FT120 (FOSS Electric, Hillerød, Denmark), content of casein with the Perov’s method modified by Tchuryna [23], profile of fatty acids with the gas chromatography method on a gas chromatograph Pye Unicam P4 4600 (Pye Unicam, Cambridge, UK) with an FID detector and using a capillary column CP-Sil 88 of a length of 50 m × 0.25 mm i.d. × 0.20 μm film; temperature of the detector 250 °C, column 180 °C; injector 250 °C; dosing a sample—split/splitless 50:1; helium was used as the carrier gas. Citric acid content was determined using a MilkoScan FT120 instrument (FOSS Electric). The analyzer employs a purpose built FTIR interferometer (Fourier Transform Infrared Spectroscopy) with the precision and stability of traditional AOAC approved fixed filter based analyzers from FOSS. The FTIR unit scans the full infrared spectrum. Identification of fatty acids was achieved according to their relative retention time. The yields of fat and protein were calculated by multiplying the yield of milk on a given day by the fat or protein content, respectively, in the milk of every cow. Milk N (kg/d) was calculated as milk true protein/6.38 + milk urea nitrogen (MUN), where milk true protein and MUN were expressed in kg/d. Milk yield (ECM) [24] was calculated as
ECM = Milk yield (kg) × [38.3 × Milk fat (g/kg) + 24.2 × Milk protein (g/kg) + 16.54 × Milk lactose (g/kg) + 20.7]/3140(1)

### 2.4. Feeds Sampling and Analysis

Representative samples of silages, concentrate, total mixed ration (TMR) and refusals were taken weekly and frozen at a temperature −25 °C. Thawed samples were dried at 60 °C in Binder dryers, and then ground in a mill (ZM 200, Retsch, Haan, Germany) to a 1 mm particle size. The proximate chemical composition of all feeds was determined by standard methods [25]. The content of DM was corrected according to the Porter and Murray method [26]. Silage samples were also assayed for pH with a pH-meter (HI 8314, Hanna Instruments, Woonsocket, RI, USA) and lactic acid by high-performance liquid chromatography (HPLC SHIMADZU) in a MetaCarb 67H P/N 5244 column (Varian, Palo Alto, CA, USA) and 0.0025 M sulfuric acid as the mobile phase, according to the manufacturer’s protocol. VFAs concentrations were determined using a gas chromatographer Varian 450-GC coupled with a flame ionization detector (FID) and a 25-meter-long capillary column CP-FFAP (the internal diameter was 0.53 mm and the thickness of the coating film was 1.0 μm). The contents of NDF, assayed with heat-stable amylase and expressed exclusive of residual ash, acid detergent fiber (ADF), expressed exclusive of residual ash, and acid detergent lignin (ADL) was determined by the method proposed by Van Soest et al. [27], using an ANKOM220 fiber analyzer (ANKOM Technology Corp., Macedon, NY, USA). The content of ammoniacal nitrogen (N-NH_3_) were determined by direct distillation in a 2100 Kjeltec Distillation unit (FOSS Analytical A/S, Hillerød, Denmark) after increasing the pH of the samples by adding MgO. N protein was determined with the help of trichloroacetic acid (TCA) [28]. The content of nonprotein nitrogen (NPN) was determined as N total – N protein. In order to estimate the nutritional value of each silage, its in vitro DM digestibility (IVDMD) was determined by keeping a silage sample in a Daisy II incubator (ANKOM Technology Corp., Macedon, NY, USA) for 48 h with the ruminal fluid, after which extraction in neutral detergent solution was carried out according to the method proposed by Kowalski et al. [29]. The nutritional value of the silage was finally estimated with the help of the WINWAR software [30]. The chemical composition of the silages is specified in Table 2. The fatty acid composition was determined by gas chromatography (GC) on a VARIAN CP-3800 chromatograph (Varian Analytical Instruments 2700 Mitchell Drive, Walnut Creek, CA 94598-1675, USA) equipped with a flame ionization detector (FID). Fatty Acid Methyl Esters (FAMEs) were prepared by the Peisker’s method, as modified by Żegarska et al. [31]. The FAMEs were separated using a CP Sil 88 (0.20 μm) capillary column 50 m long and 0.25 mm in diameter. The column temperature was 180 °C. Injector and detector temperatures were both 250 °C. Helium carrier gas flow was 1.2 mL/min at a split ratio of 1:50. The composition of fatty acids in the silages is presented in Table 3.

Apparent total-tract digestibility of DM, CP, NDF, ADF was measured after 20 d of each period, using acid-insoluble ash (AIA) as an internal marker [32]. For 5 days, samples of feeds, refusals and feces were collected and frozen at a temperature of −25 °C. Fecal samples weighing approximately 100 g each were collected for all each cow from all excreta evacuated between 7:00 a.m. and 2:00 p.m. This schedule provided five representative samples of feces for each animal. After thawing, all samples from 5 days from a cow were mixed and blended, and then a laboratory sample was taken for determination of N total with the Kjeldahl method. The remaining mass was dried at 60 °C for 72 h, ground to pass a 1-mm screen, and stored for other chemical analyses. Digestibility was calculated from the intake and concentrations of AIA, as well as these components in the diets fed, refusals and in feces, according to this equation:Intake of digested nutrient (kg/d) = intake of digested nutrient (kg/d) × {100 − [100 × (AIAd/AIAf) × (nutrientf/nutrientd)]}(2)
where AIAd = AIA concentration in the diet actually consumed, AIAf = AIA concentration in the feces, nutrientf = concentration of the nutrient in the feces, and nutrientd = concentration of the nutrient in the diet actually consumed [33,34].

### 2.5. Ruminal Fermentation Characteristics

The tests on ruminal fermentation were carried out at the Lipowo Experimental Station, affiliated with the University of Warmia and Mazury in Olsztyn, on three cannulated heifers, on days 15, 16 and 17 of feeding the experimental diet. Every heifer received each experimental diet. Samples weighing 250 g were collected at 0, 2, 4 and 6 h after feeding, from different areas in the rumen. Two separate 5-mL samples were immediately frozen for analyses of N-NH_3_ and VFA. The concentration of N-NH_3_ in the rumen contents was determined by direct distillation on a 2100 Kjeltec Distillation unit (Foss Analytical A/S, Hilleröd, Denmark) after increasing the pH of the samples by adding MgO. The pH was measured with a pH-meter (HI 8314, Hanna Instruments, Woonsocket, RI, USA). To determine VFA, 5-mL samples were collected at 0 and 6 h after the morning feeding and replenished to the volume of 1 mL with 25% metaphosphoric acid, after which they were submitted to quantitative determination using GC. The separation and determination of VFAs were carried out with gas chromatography on a gas chromatographer Varian 450-GC coupled with an autosampler Varian CP-8410 and using a flame ionization detector (FID) with a capillary column CP-FFAP length of 25 m (inner diameter 0.53 mm, film thickness 1.0 μm). The amount of a sample placed in the chromatographer was 1 μL. The temperature of the detector was 260 °C, that of the injector was 200 °C and that of the column 90 °C at the beginning of the analysis and 200 °C at the end of the analysis. The analysis parameters in the GC column were as follows: 90 °C–1.3 min; 105 °C, 25 °C/min–1.6 min; 190 °C, 25 °C/min–4.1 min, 200 °C, 25 °C/min–5.6 min. Helium was the carrier gas (flow at 5.0 mL/min) [35].

### 2.6. Statistical Analysis

Nutrient data were collated for each cow according to each measurement period, providing nine samples for analysis per period. All data were processed with the help of Statistica (Statsoft, version 13.1, TIBCO Software Inc., Palo Alto, CA, USA). Data for intake, digestibility, milk production, efficiency and composition were transformed into weekly averages for each cow by measurement period and analyzed with the aid of a model that included the effects of dietary treatment and group (square). Cow, period, and cow by period by group were the terms of the random statement. Data for ruminal pH, VFA and N-NH_3_ were analyzed with a model that included the dietary treatment. Cow and period were the terms of the random statement. Means were compared using a post hoc Duncan’s test. Unless otherwise stated, significance was declared at *p* ≤ 0.05, and tendency toward significance at 0.05 < *p* ≤ 0.10.

## 3. Results

### 3.1. Ruminal Fermentation Characteristics

Partial or complete substitution of alfalfa with Virginia fanpetals caused an increase in the sum of VFA and acetic acid (*p* < 0.01); in turn, complete substitution of alfalfa with Virginia fanpetals resulted in an increase of the A/P ratio (*p* < 0.01), higher content of N-NH_3_ in N (*p* < 0.01) and a decrease in % share of propionic acid in total VFA (Ʃ VFA) (Table 4). The Virginia fanpetals 100 diet also caused a higher % share of acetic acid in comparison with the Alfalfa 100 group. The highest sum of VFA, including acetic acid and propionic acid, as well as the lowest share of N-NH_3_ in total N were observed in the ruminal contents of cows from the group Alfalfa 50:Virginia fanpetals 50.

### 3.2. Animal Performance

A statistical analysis confirmed differences in the intake of DM, CP, NDF, ADF and in milk yield/DMI between groups (Table 5). Cows fed the diet without alfalfa silage (Virginia fanpetals 100) had a higher intake of CP and NDF (*p* < 0.01) than cows eating diets with alfalfa silage. Regarding DM, the statistical analysis demonstrated its higher intake in the Virginia fanpetals 100 group than in Alafalfa 100 one. The addition of Virginia fanpetals silage to feeding rations resulted in a lower intake of ADF. In turn, complete substitution of alfalfa silage with Virginia fanpetals silage caused a decrease of the milk yield/DMI ratio (*p* < 0.01). The inclusion of Virginia fanpetals silage in two feeding rations did not affect the digestibility of nutrients from the rations, nor did it influence the milk yield or the content of protein and fat in milk.

### 3.3. Milk Yield and Composition

The substitution of alfalfa silage with Virginia fanpetals silage did not cause many changes in the chemical composition of milk (Table 6). The milk from the Alfalfa 50:Virginia fanpetals 50 cows was characterized by a higher content of protein (*p* = 0.03) and a tendency towards an increased casein content in comparison with the milk from the other groups of cows. In turn, the milk from the Virginia fanpetals 100 group contained more citric acid and urea (*p* = 0.04) than the milk from the Alfalfa 100 group. The remaining differences were not confirmed statistically.

### 3.4. Profile of Milk Fatty Acids

A statistical analysis confirmed differences in the composition of fatty acids in milk (*p* < 0.01) (Table 7). The inclusion of Virginia fanpetals silages in the diets Alfalfa 50:Virginia fanpetals 50 and Virginia fanpetals 100 resulted in a higher ratio of saturated fatty acids to unsaturated ones, including monounsaturated fatty acids (MUFA). Milk from these two groups was also observed to have a lower content of n-6 acids, and therefore, a decrease in the MUFA/SFA ratio followed. With respect to polyunsaturated fatty acids (PUFA) and n-3 acids, their decrease was noted only in the group fed a ration without alfalfa silage (Virginia fanpetals 100). Milk from this group had a n-6/n-3 ratio higher than milk from the other cows.

### 3.5. Functional Fatty Acids in Milk

As the share of Virginia fanpetals silage in a feeding ration increased, so did the content of vaccenic acid (*p* < 0.01; Table 8). Milk from the cows fed Virginia fanpetals silage contained less oleic and linoleic acids (both at *p* < 0.01). Milk from the cows fed a feeding ration without alfalfa silage (Virginia fanpetals 100) contained less linoleic acid than milk from groups receiving diets with alfalfa silage (*p* < 0.01; Alfalfa 50:Virginia fanpetals 50 and Alfalfa 100). Milk from the former group also contained less arachidonic acid (*p* = 0.04) and showed a tendency towards a smaller amount of docosapentaenoic acid (DPA) in comparison with milk from the Alfalfa 100 group. Generally, the analytical results on functional fatty acids in the Alfalfa 50:Virginia fanpetals 50 were between those from the other groups. The quantities of eicosapentaenoic acid (EPA) and docosahexaenoic acid (DHA) were similar in milk from all the groups, and differences in this regard were not confirmed statistically.

## 4. Discussion

Chemical analyses of the silages (Table 1) confirmed the value of Virginia fanpetals silage, with CP contents similar to those observed with alfalfa silage. Virginia fanpetals silage is characterized by lower content of NDF, ADF and nearly times lower ADL content in NDF (ADL/NDF) (7.25% vs. 13.4%). Both protein silages were well preserved, and the quality of fermentation was very good [36]. Some weak signs of secondary fermentation in Virginia fanpetals silage were a consequence of direct harvest without previous wilting [37]. This ensiling technology was adopted after preliminary unsuccessful wilting, when the cut biomass was observed to heat on swaths. In turn, shaking and raking might have caused large losses of protein due to thick stems and delicate leaves drying at different rates. It was decided not to use a silage inoculant, as our earlier studies had demonstrated a good fermentation profile achieved without any ensiling additives [38].

According to the cattle nutrition standards [22], the experimental rations satisfied the requirements set for NE and protein digested in the small intestine (PDI) for cows producing 26 kg milk/daily containing 4.4% of fat; however, the ration supplied to the control group contained less PDI by 2 g. The higher supply of the groups fed Virginia fanpetals silage with energy and protein was a consequence of the higher intake of DM, which was confirmed by the lower fill unit value of this silage compared to alfalfa silage. The higher intake of diets including Virginia fanpetals silage could be explained by the their IVDMD, which was higher than in alfalfa silage [39]. The higher share of butyric acid and ammoniacal nitrogen in Virginia fanpetals silage did not have a negative effect on the intake of the diets. The content of ammoniacal nitrogen in Virginia fanpetals silage did not exceed 100 g/kg N, which is considered to be an acceptable amount [36]. In a study by Dewhurst et al. [39], moderate amounts of butyric acid in red clover silage did not preclude the high intake of rations and high milk yield. Although the intake of diets with Virginia fanpetals silage was higher, the milk yield was similar in all the groups, which implicates the lower utilization of nutrients. The influence should be noted of a small share of straw in the Alfalfa 50:Virginia fanpetals 50 and Virginia fanpetals 100 groups. The decrease in the DM utilization rate may have been the result of introducing a small amount of straw to the groups. Similarly, in studies by Ferris et al. [40], the inclusion of a small amount of straw in the diets of dairy cows resulted in an increase DM intake while lowering feed conversion. The intake of digestible NDF in the experimental groups was ca. 4.35, 4.31 and 4.38 kg/d, respectively. A high percentage of small particles causes the rapid passage of the gastric contents through the rumen, which facilitates a high intake of feeds; on the other hand, the digestibility and milk yield are lower than one might expect in view of the high feed intake [41]. In this experiment, feeding cows with Virginia fanpetals silage led to a higher production of VFA, which suggests enhanced ruminal fermentability stimulated by the higher feed intake. In this group, too, the percentage of propionic acid in total VFA was the lowest, while that of acetic acid was the highest, which manifested in worse conversion of the feed and led to a slightly lower milk yield with a lower content of UFA acids and functional acids. Changes in proportions among VFA, especially the increase in propionic acid and decrease in acetic acid, are often observed as a result of enhanced digestion of fiber in the rumen, which may induce a rise in counts of microorganisms, shifts in metabolic pathways, better conversion of nutrients, higher milk yield and higher fat content of milk [33,42]. In this study, a decrease in the share of propionic acid and an increase in acetic acid, as well as slightly lower milk production parameters, were observed. Berthiaume et al. [43] studied high and low NDF digestibility alfalfa varieties in vitro and noticed enhanced total production of VFA, as well as higher apparent digestibility of dry matter. The concentrations of VFA and acetic acid in rations with Virginia fanpetals may be attributed to the higher IVDMD in Virginia fanpetals silage, caused by the lower ADL/NDF and higher contribution of hemicellulose in NDF. The previously noted relationship between the higher content of total VFA and concurrently lower pH in the rumen contents [44] was confirmed. The content of NDF in diets supplied to all the nutritional groups proved to be sufficient to maintain the optimal ruminal pH. A slight decline in pH was recorded in the groups fed Virginia fanpetals silage, which was a result of the higher production of acetate and propionate. The production of acids was modified by the greater fragmentation of particles and higher moisture content in Virginia fanpetals silage. These factors may have translated into a shorter chewing time, lower production of saliva and shorter retention of the contents in the rumen. However, the slight decrease in pH turned out to be irrelevant, as proved by the good milk yield and high fat content in milk [45].

The ratio of CP digestibility to DM digestibility (CPD/DMD) was 1.12, 1.13 and 1.05, respectively, which suggests poorer digestibility of protein from the rations composed of Virginia fanpetals silage fed exclusively with maize [39]. As a result, the intake of digestible protein was, respectively, 2.27, 2.26 and 2.24 kg/d. The worse utilization of nitrogen in the Virginia fanpetals 100 group of cows may have been due to higher losses in the rumen resulting from the higher share of NPN in total N compared to alfalfa silage. This confirms a significantly higher concentration of N-NH_3_ and milk urea nitrogen (MUN) in the rumen contents in the Virginia fanpetals 100 group. An increase in VFA in the rumen contents was not high enough to offset the increase in N-NH_3_ induced by the complete substitution of alfalfa silage with Virginia fanpetals silage. Depressed efficiency of the bacterial protein synthesis as a result of the increased NPN in silage was confirmed by Dewhurst et al. [39] and Winters et al. [46]. Nevertheless, the nitrogen conversion ratio for milk protein synthesis in all dietary treatments should be deemed as high, which may have been a result of the inclusion of maize silage in the feeding rations, and might indicate a balance between easily available N and digestible carbohydrates. Dewhurst et al. [39] reported that cows fed alfalfa silage supplemented with a concentrate demonstrated N efficiency at 0.182. Improved utilization of nitrogen in the rumen achieved by microbiota occurs in rations based on maize silages owing to the increased contribution of starch in the diet [47]. The efficiency of N utilization from cows’ diets based on silage made from maize harvested at different maturity stages ranged from 0.299 to 0.323 [48], which was associated with the decreasing amounts of N discharged with urine as a result of a higher starch intake. The tendency towards an increase in the total VFA concentration in the rumen due to the inclusion of Virginia fanpetals silage in the feeding ration proves that the mixture of alfalfa silage and Virginia fanpetals silage achieved the best fermentability in the rumen, which may have favored a better supply of energy for lactation. This effect could have partly been caused by the composition of fiber and the flow rate of particles. The higher share of propionate relative to acetate can be perceived as a direct effect of the increased fiber digestion in the rumen [42]. Propionate is quantitatively the major precursor of glucose; hence, it has a strong impact on the release of hormones and distribution of nutrients into tissues [49]. In consequence, the elevated VFA concentration and higher proportion of propionate in the Alfalfa 50:Virginia fanpetals 50 group improved the supply of glucose and production of milk components. Better NDF digestibility, confirmed by the higher VFA concentration, was able to ensure a higher energy supply and better bacterial protein production efficiency. In in vivo studies on high and low NDF digestibility alfalfa varieties, an increase in the total VFA production and increased apparent DM and OM digestibility, contributing to a rise in microbial crude protein, were observed [43]. The tendency towards better utilization of total N and digestible N in this group can also be attributed to the mutually supplementing composition of amino acids of the proteins from the silages [50]. Our unpublished results from a study into the composition of Virginia fanpetals silage proteins showed that in comparison with alfalfa silage, the CP of Virginia fanpetals silage had a higher content of essential amino acids Lys, Thr, the content of which was as follows: (3.98 vs. 2.62), (4.19 vs. 2.14) g/100 g CP. This assumption was supported by the fact that better N efficiency coincided with better digestible N efficiency, which equaled 0.39, 0.41 and 0.39, in the respective three dietary groups.

The effect of Virginia fanpetals silage on the chemical composition of milk was quite weak. Feeding a mixture of Virginia fanpetals silage with alfalfa silage slightly raised the total protein content, which most probably was a consequence of the higher share of urea and a tendency towards a higher content of casein. In the same dietary group, nitrogen utilization was minimally higher, but this was not verified statistically. The fat content in milk was high in all groups, which could have resulted from the high share of acetate in VFA, typical of a high forage diet NDF in feeding rations [33]. The remaining milk parameters were characteristic for normal milk.

As for the composition of fatty acids, Virginia fanpetals silage provided a similar amount of SFA as alfalfa silage, which ensured less MUFA and more PUFA. Virginia fanpetals silage was also determined to contain more lauric, stearic and linoleic acids, but less margaric and oleic acids. Virginia fanpetals silage modified the profile of milk fatty acids by slightly increasing the content of saturated fatty acids, which is unfavorable to human health. The most beneficial profile of particular groups of fatty acids was identified in milk from the cows fed a ration without Virginia fanpetals silage, whereas milk from the other groups, receiving Virginia fanpetals silage alone or in a mixture with alfalfa silage, was characterized by similar composition of fatty acids in individual categories of these compounds. There are reports in the literature suggesting that rations based on legume silages (alfalfa, white clover, red clover) cause a decrease in the milk fat content and an increase in PUFA, n-6 and n-3 acids in comparison to grass silages [51]. The results obtained in this study are rather contrary, and can be compared to those observed when cows are fed rations containing grass silages [52]. This can be explained by the dilution effect, as the milk fat content was high, or by the aforementioned rapid passage of gastric contents and shorter retention time in the rumen [51]. The detailed analysis of functional fatty acids in milk showed quite small differences induced by the inclusion of Virginia fanpetals silage to rations. The biggest differences were noted for oleic and vaccenic acids. Oleic acid, as a representative of long-chain fatty acids, is released in milk due to its intake with fodder. Virginia fanpetals silage had a much lower content of oleic acid than that found in alfalfa silage, which resulted in smaller concentrations of this acid in milk; this was not compensated for by the higher DMI in the Virginia fanpetals 100 group [53]. In turn, vaccenic acid in milk is a product of the biohydrogenation of linoleic and linolenic acids in the rumen, and its amount is strictly coupled with CLAs [53]. Regarding functional fatty acids, differences resulting from feeding cows with alfalfa silage or Virginia fanpetals silage are small. This may be attributed to considerable similarity between alfalfa and Virginia fanpetals silages, both in terms of their fatty acid composition and in their chemical composition in general. In other studies where alfalfa silage was used, a similar functional acid profile was obtained to that of Alfalfa 100, especially in CLA, LNA, AA, EPA, DPA and DHA [54].

## 5. Conclusions

This research confirmed that it is possible to include biomass of Virginia fanpetals harvested at the early vegetation stage in rations fed to dairy cows. Virginia fanpetals silage did not affect the digestibility of nutrients in rations, nor did it influence milk yield. Complete substitution caused an increase in DMI, total VFA, A/P ratio and N-NH_3_ content in the rumen contents and urea in milk, and a decrease in the feed conversion ratio. Both partial and complete substitution of alfalfa with Virginia fanpetals changed the profile of fatty acids, resulting in a slight increase in SFA and a decrease in UFA. The content of all functional fatty acids except vaccenic acid declined. The determined high potential of the intake of rations and milk production suggests that Virginia fanpetals and maize may complement each other in regions where soils are characterized by low suitability for agricultural use. However, in this study, Virginia fanpetals silage did not significantly improve animal performance and diminished milk quality.

## Figures and Tables

**Table 1 animals-10-01746-t001:** Ingredients and chemical composition of experimental diets fed to dairy cows (*n* = 3 sample replicates).

Specification	Feeding Ration ^+^		
	Alfalfa 100	Alfalfa 50:Virginia Fanpetals 50	Virginia Fanpetals 100
Composition of a ration, % DM			
Maize silage	35	35	34
Virginia fanpetals silage	-	17	34
Alfalfa silage	35	17	-
Triticale straw	-	1	2
Corn grain	7.8	7.8	7.8
Triticale	7	7	7
Beet pulp	4	4	4
DDGS	3	3	3
RSM	3	3	3
Soya meal	3	3	3
Calcium carbonate	1.1	1.1	1.1
Salt	0.35	0.35	0.35
Mineral vitamins ^++^	0.15	0.15	0.15
Sodium bicarbonate	0.6	0.6	0.6
DM (g/kg FM) in DM g/kg	428.8	353.7	301.3
Organic matter (OM)	921.6	921.7	927.8
Crude protein (CP)	147.7	147.0	147.5
NDF	378.8	379.8	381.0
ADF	248.7	239.2	231.5
UFL	0.91	0.92	0.93
PDIN	92	92	93
PDIE	87	89	91

^+^ Alfalfa 100, diets without Virginia fanpetals silage; Alfalfa 50:Virginia fanpetals 50, diets with alfalfa silage and Virginia fanpetals silage; Virginia fanpetals 100, diets without alfalfa silage; ^++^ Containing: calcium carbonate 15.95%, sodium chloride 10%, magnesium oxide 5%, natrium-calcic phosphate and phosphate 1-Ca 4%, Mn 10,400 mg, Zn 19,500 mg, Cu 3450 mg, I 400 mg, Se 120 mg; Co 100 mg, vitamin A 2,000,000 IU; vitamin D 200,000 IU; vitamin E 12.000 mg; vitamins of group B 20.750 mg per kg; NDF, neutral detergent fiber; ADF, acid detergent fiber; UFL, feed unit for milk production; PDIN, protein truly digestible in the small intestine when N limits microbial protein synthesis; PDIE, protein truly digestible in the small intestine when energy limits microbial protein synthesis.

**Table 2 animals-10-01746-t002:** Chemical composition of silages (g/kg DM).

Specification	Silage			SEM ^+^
	Maize	Alfalfa	Virginia Fanpetals	
Dry matter (g/kg FM)	323.9	397.3	177.3	
in DM g/kg				
Organic matter (OM)	962.2	895.0	898.6	4.38
Crude protein (CP)	78.6	185.0	190.3	1.92
Ether extract (EE)	26.1	21.1	20.7	1.02
NDF	432.8	451.7	437.1	7.49
ADF	252.2	389.8	327.1	8.24
ADL	19.7	40.6	31.7	1.56
pH	3.84	4.54	4.46	0.05
Lactic acid	48.9	44.3	78.4	2.37
Acetic + propionic acid	5.21	7.73	26.6	6.19
Butyric acid	0.02	0.24	6.72	0.76
N-NH_3_ g/kg N	31.1	66.5	98.5	14.7
NPN g/kg N	ND	558.5	713.4	11.4
IVDMD	710	610	650	0.04
UFL	0.90	0.77	0.87	0.02
PDIN	50	110	118	4.3
PDIE	67	79	91	3.6
FUC	1.04	1.10	1.03	0.02

^+^ SEM, Standard error of mean; FM, fresh matter; NDF, neutral detergent fiber; ADF, acid detergent fiber; ADL, acid detergent lignin; N-NH_3_, ammoniacal nitrogen; NPN, non-protein nitrogen; IVDMD, in vitro dry matter digestibility; UFL, feed unit for lactation; PDIN, protein digested in the small intestine; PDIE, protein digested in the small intestine calculated according to the feed energy available in the rumen; FUC, fill unit for dairy cows; ND, non determined.

**Table 3 animals-10-01746-t003:** Composition of fatty acids in the silages, expressed in g/100 g sum of acids.

Acids	Silage		SEM ^+^
	Alfalfa	Virginia Fanpetals	
C12:0 Lauric acid	0.21	0.54	0.02
C12:1 Lauroleic acid	1.47	1.27	0.04
C14:0 Myristic acid	1.09	1.21	0.11
C14:1 Myristoleic acid	0.30	0.32	0.01
C15:0 Pentadecyclic acid	1.18	1.25	0.09
C16:0 Palmitic acid	28.61	27.00	6.43
C16:1 Palmioleic acid	1.23	1.25	0.12
C17:0 Margaric acid	0.84	0.52	0.03
C17:1 Margaoleic acid	0.47	0.41	0.05
C18:0 Stearic acid	3.72	6.04	0.23
C18:1 c9 oleic acid	11.31	5.81	1.08
C18:2 Linoleic acid	20.31	27.39	3.78
C18:3 α-linolenic acid	29.23	26.99	7.13
SFA	35.66	36.56	0.68
MUFA	14.79	9.06	1.30
PUFA	49.55	54.38	1.32

^+^ SEM, standard error of mean; SFA, saturated fatty acids; MUFA, monounsaturated fatty acids; PUFA, polyunsaturated fatty acids.

**Table 4 animals-10-01746-t004:** Ruminal fermentation characteristics of lactating dairy cows fed different silage-diets.

Specification	Feeding Ration ^+^			SEM ^++^	*p*
	Alfalfa 100	Alfalfa 50:Virginia Fanpetals 50	Virginia Fanpetals 100		
pH					
0 h	6.52	6.53	6.55	0.067	0.99
2 h	6.35	6.11	6.28	0.068	0.37
4 h	6.37	5.82	6.05	0.110	0.11
6 h	6.14	5.74	5.88	0.097	0.26
N-NH_3_, g/100g N	5.00 ^a^	4.46 ^a^	6.95 ^b^	0.350	<0.01
Acetic acid, mmol	20.17 ^a^	26.86 ^b^	25.10 ^b^	0.508	<0.01
Propionic acid, mmol	3.14 ^a^	4.24 ^b^	3.47 ^a^	0.145	<0.01
Butyric acid, mmol	1.29	1.48	1.59	0.071	0.21
Total VFA, mmol	24.60 ^a^	32.58 ^b^	30.16 ^b^	0.997	<0.01
A/P	6.42 ^a^	6.34 ^a^	7.23 ^b^	0.092	<0.01
A, (%)	82.00 ^a^	82.45	83.22 ^b^	0.183	0.03
P, (%)	12.76 ^a^	13.01 ^a^	11.51 ^b^	0.145	<0.01
B, (%)	5.24	4.54	5.27	0.150	0.17

^+^ Alfalfa 100, diets without Virginia fanpetals silage; Alfalfa 50:Virginia fanpetals 50, diets with alfalfa silage and Virginia fanpetals silage; Virginia fanpetals 100, diets without alfalfa silage; ^++^ SEM, standard error of mean; N-NH_3_, ammoniacal nitrogen; VFA, volatile fatty acids; A, acetic acid; P, propionic acid; B, butyric acid; a, b, *p* ≤ 0.05.

**Table 5 animals-10-01746-t005:** Intake, digestibility DM and nutrients, milk yield and composition, and efficiencies of DM and N used for milk production of lactating dairy cows fed different silage-diets.

Specification	Feeding Ration ^+^			SEM ^++^	*p*
	Alfalfa 100	Alfalfa 50:Virginia Fanpetals 50	Virginia Fanpetals 100		
Intake, kg/d					
DM	20.0 ^a^	20.1 ^a^	20.5 ^b^	0.092	0.03
CP	2.95 ^a^	2.96 ^a^	3.03 ^b^	0.014	0.03
NDF	7.57 ^a^	7.65 ^a^	7.83 ^b^	0.036	<0.01
ADF	4.97 ^a^	4.82 ^b^	4.75 ^b^	0.023	<0.01
Digestibility, %					
DM	68.6	67.6	67.9	0.636	0.71
CP	76.9	76.4	74.1	0.669	0.14
NDF	57.5	56.4	56.0	0.737	0.66
ADF	52.5	51.7	52.7	0.684	0.85
Yield, kg/d					
Milk	25.9	26.1	25.8	0.116	0.69
ECM	28.76	30.45	28.82	0.456	0.82
Fat	1.26	1.31	1.27	0.020	0.63
Protein	0.89	0.92	0.88	0.007	0.41
Efficiency					
Milk yield/DMI	1.30 ^b^	1.30 ^b^	1.26 ^a^	0.004	<0.01
Milk N/N Intake	0.31	0.32	0.30	0.645	0.65

^+^ Alfalfa 100, diets without Virginia fanpetals silage; Alfalfa 50:Virginia fanpetals 50, diets with alfalfa silage and Virginia fanpetals silage; Virginia fanpetals 100, diets without alfalfa silage; ^++^ SEM, standard error of mean; ECM, energy-corrected milk; a, b, *p* ≤ 0.05.

**Table 6 animals-10-01746-t006:** Physicochemical parameters and selected chemical components of milk from lactating cows fed different silage-diets.

Milk Properties	Feeding Ration ^+^			SEM ^++^	*p*
	Alfalfa 100	Alfalfa 50:Virginia Fanpetals 50	Virginia Fanpetals 100		
Casein, %	2.84 ^y^	2.88 ^x^	2.84 ^y^	0.020	0.10
Density, g/cm^3^	1.031	1.031	1.031	0.158	0.91
Protein, %	3.44 ^a^	3.51 ^b^	3.44 ^a^	0.027	0.03
Fat, %	4.85	5.00	4.91	0.076	0.50
Dry matter, %	13.90	14.01	13.89	0.089	0.63
Fat-free dry matter, %	9.06	9.11	9.07	0.031	0.44
Lactose, %	4.69	4.65	4.68	0.020	0.21
Acidity	7.25	7.42	7.30	0.073	0.27
Citric acid, %	0.134	0.138	0.144	0.002	0.03
Urea, g/L	0.023 ^a^	0.023 ^b^	0.026 ^a^	0.001	0.04

^+^ Alfalfa 100, diets without Virginia fanpetals silage; Alfalfa 50:Virginia fanpetals 50, diets with alfalfa silage and Virginia fanpetals silage; Virginia fanpetals 100, diets without alfalfa silage; ^++^ SEM, standard error of mean; a, b, *p* ≤ 0.05; x, y, 0.05 < *p* ≤ 0.10.

**Table 7 animals-10-01746-t007:** Saturated, unsaturated and other groups of fatty acids in milk from lactating dairy cows fed different silage-diets.

Fatty Acids, g/100 g of Sum of Acids	Feeding Ration ^+^			SEM ^++^	*p*
	Alfalfa 100	Alfalfa 50:Virginia Fanpetals 50	Virginia Fanpetals 100		
SFA	73.86 ^a^	75.20 ^b^	75.49 ^b^	0.286	<0.01
UFA	26.14 ^a^	24.80 ^b^	24.51 ^b^	0.286	<0.01
MUFA	22.92 ^a^	21.64 ^b^	21.50 ^b^	0.260	<0.01
PUFA	3.22 ^a^	3.16 ^a^	3.01 ^b^	0.041	<0.01
PUFA/SFA	0.044 ^a^	0.042	0.040 ^b^	0.001	<0.01
MUFA/SFA	0.310 ^a^	0.289 ^b^	0.285 ^b^	0.005	<0.01
n-6	1.71 ^a^	1.63 ^b^	1.57 ^b^	0.026	<0.01
n-3	0.45 ^a^	0.44 ^a^	0.37 ^b^	0.008	<0.01
n-6/n-3	3.80 ^a^	3.71 ^a^	4.24 ^b^	0.072	<0.01

^+^ Alfalfa 100, diets without Virginia fanpetals silage; Alfalfa 50:Virginia fanpetals 50, diets with alfalfa silage and Virginia fanpetals silage; Virginia fanpetals 100, diets without alfalfa silage; ^++^ SEM, standard error of mean; SFA, saturated fatty acids; UFA, unsaturated fatty acids; MUFA, monounsaturated fatty acids; PUFA, polyunsaturated fatty acids; n-6, n-3, polyunsaturated omega fatty acids, a, b, *p* ≤ 0.05.

**Table 8 animals-10-01746-t008:** Selected functional fatty acids in milk of lactating dairy cows fed different silage-diets.

Fatty Acids, g/Sum of Acids	Feeding ration ^+^			SEM ^++^	*p*
	Alfalfa 100	Alfalfa 50:Virginia Fanpetals 50	Virginia Fanpetals 100		
C 4 (BA)	2.62	2.76	2.84	0.001	0.52
C 18:1 trans 11 (TVA)	1.32 ^a^	1.65 ^b^	1.84 ^c^	0.039	<0.01
C 18:1 cis (OA)	16.54 ^a^	15.02 ^b^	14.87 ^b^	0.244	<0.01
C 18:2 (LA)	1.61 ^a^	1.54 ^b^	1.48 ^b^	0.025	<0.01
C 18:2 cis 9 trans 11 (CLA)	0.548	0.576	0.565	0.010	0.22
C 18:3 (LNA)	0.345 ^a^	0.334 ^a^	0.275 ^b^	0.008	<0.01
C 20:4 (AA)	0.095 ^a^	0.093	0.086 ^b^	0.002	0.04
C 20:5 (EPA)	0.031	0.032	0.030	0.001	0.14
C 22:5 (DPA)	0.057 ^x^	0.056	0.051^y^	0.001	0.06
C 22:6 (DHA)	0.016	0.015	0.016	0.001	0.90

^+^ Alfalfa 100, diets without Virginia fanpetals silage; Alfalfa 50:Virginia fanpetals 50, diets with alfalfa silage and Virginia fanpetals silage; Virginia fanpetals 100, diets without alfalfa silage; ^++^ SEM, standard error of mean; BA, butyric acid; TVA, vaccenic acid; OA, oleic acid; LA, linoleic acid; CLA, conjugated linoleic acid; LNA, linolenic acid; AA, arachidonic acid; EPA, eicosapentaenoic acid; DPA, docosapentaenoic acid; DHA, docosahexaenoic acid; a, b, *p* ≤ 0.05; x, y, 0.05 ≤ p ≤ 0.1

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
