# Peer review of "Effects of Dietary Substitution of Alfalfa Silage with Virginia Fanpetals Silage in Lactating Polish Holstein Friesian Dairy Cows"

_animals, 2020, doi:10.3390/ani10101746_

Round 1

Reviewer 1 Report

The paper have two aims: confirming whether it is feasible to use biomass of Virginia fanpetals in nutrition of dairy cows, and to determine the effect of partial or complete substitution of alfalfa silage with Sida silage in nutrition of lactating Polish Holstein Fresian cows on the intake and digestibility of rations, ruminal fermentation, milk yields as well as the physicochemical parameters of milk and the composition of fatty acids in milk. Both objectives could be summarised on a single objective.

The study is within the scope of Animals. The authors did a good work from an experimental point of view. The study is original and interesting, however its implementation is limited in dairy farming because the improvement is marginal with regard to alfalfa silage. It would have been interesting to include a cost-benefit analysis.

The wording is unclear because some paragraphs and tables are disordered and mixing sections (see specific comments).

Please look at my suggestions and specific comments in the attached file.

Author Response

We would like to thank the Reviewer for a thorough persual of the manuscript and valuable comments and suggestions which have enabled us to improve its quality. We are sending responses to the Reviewer's comments and suggestions in the attachment. 

Reviewer 2 Report

Title

The title is too long

Introduction

Line 61: Please revise “Virginia fanpetals (Sida hermaphrodita R.)”; it needs to be completed: L. Rusby.

Line 65: Consider to use “cutting interval” instead of “cutting time”

Lines 75 – 77. The sentence needs a reference

It needs to be consistent when referring to a botanical name. The first time it is mentioned, it should be Virginia fanpetals (Sida hermaphrodita L. Rusby), after you should use the genus abbreviation and the full species name (S. hermaphrodita), or the common name, Virginia fanpetals.

Table 2: Title needs to be completed.

Line 178: Please indicate the meaning of MUN

Line 179: Clarify “Milk yield (ECM)”, do you mean Fat-corrected milk yield (FCM) or energy-corrected milk (ECM)? In Table 5 you indicated: “ECM, 4% fat-corrected milk”.

FCM = 0.4 × Milk yield (kg/d) + 15 × Milk fat (kg/d) (NRC, 2001).

ECM = Milk yield (kg) × [38.3 × Milk fat (g/kg) + 24.2 × Milk protein (g/kg) + 16.54 × Milk lactose (g/kg) + 20.7]/3140 (Schau and Fet, 2008)

1) Do you use the above formulae to calculate fat-corrected milk or energy-corrected milk?  

2) The reference cited (J. Rezaei et al. 2015) is not the original (Schau and Fet, 2008) source for this equation to calculate ECM

269: Ʃ VFA do you mean …total VFA?

279: To correct, it should de Table 6

280: To correct, it should be Table 7

291: To correct, it should be Table 8

305: To correct, it should be Table 9

Lines 305-306: Please revise the phrase: “Milk from the cows fed Sida silage contained more oleic acid and less linoleic acid (both at P<0.01)”; it does not agree with Table 9.

Line 308 “Milk from the former group also contained less arachidonic acid ….” In table 9 you reported “AA, arachidic acid” Please clarify.

DISCUSION

321: Redaction needs to be improved

321-322 and 323 Use “Sida silage”  or “Virginia fanpetals”, but not both

329 Meaning of DLG

Author Response

(The authors gave the same response as above.)

Reviewer 3 Report

Ms. Ref. No.: animals-914219

Title: “Effects of dietary substitution of alfalfa silage with Virginia fanpetals (Sida hermaphrodita) silage on feed intake, digestibility, ruminal fermentation, milk yield, fatty acids composition and nitrogen retention in lactating Polish Holstein Fresian cows”

General comments

I have had the opportunity to review the manuscript. The manuscript is interesting and is in the topic of the journal, and can make a positive contribution to the scientific community

Below my considerations:

Title:

The title is too long, I suggest to modify it as follows:

“Effects of dietary substitution of alfalfa silage with Virginia fanpetals silage in lactating Polish Holstein Fresian dairy cows”

Introduction

The introduction is well described and structured, in some aspects it could also be reduced, see lines 75-77 deleted or add the references.

Material and methods

L 128 add unit of measure

Table 1 add if there are significant differences and possibly include them in discussions

L 224 aggiungere la bibliografia di riferimento della metodologia utilizzata:

Van Keulen, J., Young, B.A., 1977. Evaluation of acid-insoluble ash as a nat-ural marker in ruminant digestibility studies. J. Anim. Sci. 44, 282–287, and also:

In vivo digestibility of two different forage species inoculated with arbuscular mycorrhiza in Mediterranean red goats. Small Rumin. Res. 123, 83-87.

Results

Table 5 DM 20.1 add letter (a)

Discussion

The discussions are well structured, it is suggested to identify some comparisons with other species and to evaluate possible effects on the sensory analysis of milk.

Author Response

(The authors gave the same response as above.)

Round 2

Reviewer 1 Report

This is the second round of review of this paper. The authors have improved significantly the manuscript. However, there is a point of attention that need to be addressed before publishing. For the sake of reproducibility, I request that the methods, which are the most important for the results, are presented in the materials. For this reason, the authors must explicit the rate of temperature applied on the GC column (L239).

Author Response

We would like to thank the Reviewer for a thorough persual of the manuscript and valuable comments and suggestions which have enabled us to improve its quality. The answer is in the attachment. 
